

# The impact of the ozone effective temperature on satellite validation using the Dobson spectrophotometer network

M. E. Koukouli[1], M. Zara[1,a], C. Lerot[2], K. Fragkos[1], D. S. Balis[1], and M. van Roozendael[2]

[1]Laboratory of Atmospheric Physics, Aristotle University of Thessaloniki, Thessaloniki, Greece
[2]Belgian Institute for Space Aeronomy (BIRA-IASB), Brussels, Belgium
[a]now at: Koninklijk Nederlands Meteorologisch Instituut (KNMI), De Bilt, the Netherlands

Received: 9 November 2015 – Accepted: 10 December 2015 – Published: 14 January 2016

Correspondence to: M. E. Koukouli (mariliza@auth.gr)

Published by Copernicus Publications on behalf of the European Geosciences Union.

Discussion Paper | Discussion Paper | Discussion Paper | Discussion Paper |

**AMTD**

doi:10.5194/amt-2015-343

The impact of the ozone effective temperature

M. E. Koukouli et al.



## Abstract

The main aim of the paper is to demonstrate an approach for post-processing of the Dobson spectrophotometers total ozone columns [TOCs] in order to compensate for their known stratospheric effective temperature ($T_{eff}$) dependency and its resulting ef-
⁵ fect on the usage of the Dobson TOCs for satellite TOCs validation. The Dobson observations employed are those routinely submitted to the World Ozone and UV Data Centre (WOUDC) of the World Meteorological Organization whereas the effective temperatures have been extracted from two sources: the European Space Agency, ESA, Ozone Climate Change Initiative, Ozone-CCI, GODFIT version 3 (GOME-type
¹⁰ Direct FITting) algorithm applied to the GOME2/MetopA, *GOME2A*, observations as well as the one derived from the European Centre for Medium-Range Weather Forecasts (ECMWF) outputs. Both temperature sources are evaluated utilizing co-located Ozonesonde measurements also retrieved from the WOUDC database. Both GOD-FIT_v3 and ECMWF $T_{eff}$s are found to be unbiased against the ozonesonde observa-
¹⁵ tions and to agree with high correlation coefficients, especially for latitudes characterized by high seasonal variability in $T_{eff}$.

The validation analysis shows that, when applying the GODFIT_v3 effective temperatures in order to post-process the Dobson TOC, the mean difference between Dobson and GOME2A GODFIT_v3 TOCs moves from $0.63 \pm 0.66$ to $0.26 \pm 0.46$ % in the North-
²⁰ ern Hemisphere and from $1.25 \pm 1.20$ to $0.80 \pm 0.71$ % in the Southern Hemisphere. The existing solar zenith angle dependency of the differences has been smoothed out, with near-zero dependency up to the 60 to 65° bin and the highest deviation decreasing from $2.38 \pm 6.6$ to $1.37 \pm 6.4$ % for the 80 to 85° bin. We conclude that the global scale validation of satellite TOCs against collocated Dobson measurements benefits
²⁵ from a post-correction using suitably estimated $T_{eff}$s.

Discussion Paper | Discussion Paper | Discussion Paper | Discussion Paper |

# AMTD

doi:10.5194/amt-2015-343

**The impact of the ozone effective temperature**

M. E. Koukouli et al.

# 1 Introduction

Satellite observations of the total ozone column (hereafter, TOC) on a global scale have routinely been performed since the early 1980s and in the later years even concurrently by multiple instruments on different polar platforms such as the TOMS/EP, GOME/ERS-2, SCIAMACHY/Envisat, OMI/Aura and the recent OMPS/Suomi NPP, among others. The validation of these measurements using ground-based instrumentation as "truth" has also been an integral part of the satellite TOC time series production. Since year 1958, also known as the International Geophysical Year, when the need for routine global TOC measurements was clearly demonstrated, the first world-wide network of manually operated Dobson spectrophotometers was established. Later on, in the early 1980s, the fully automated Brewer spectrophotometer was launched and the global monitoring of the atmospheric ozone content was thus enhanced. Innumerous satellite validation studies have used these ground-based observations in order to assess the behaviour and accuracy of both their measurements and algorithm (for e.g. Lambert et al., 1999; Fioletov et al., 1999; Lambert et al., 2000; Bramstedt et al., 2003; Weber et al., 2005; Balis et al., 2007a; among others.) As satellite instrumentation technology advanced and the associated retrieval algorithms became more sophisticated the unavoidable shortcomings of the ground-based measurements became more of an issue than before. One such concern is the fact that the operational Dobson algorithm does not account for the natural intra-annual variability of the stratospheric temperature which in turn heavily affects the ozone absorption coefficients used in the Dobson TOC retrieval. This algorithmic short-coming results in seasonal ozone column dependencies being introduced which hinders the real performance of satellite total ozone algorithms when validated with Dobson measurements.

In this paper we shall introduce a post-processing of the daily TOC values formally reported to the World Meteorological Organization (WMO) World Ozone and UV Data Centre (WOUDC) database. Effective temperatures, i.e the weighting of the atmospheric temperature profile with the ozone profile, hereafter $T_{eff}$, from both an algorithm

## AMTD

doi:10.5194/amt-2015-343

### The impact of the ozone effective temperature

M. E. Koukouli et al.

Discussion Paper | Discussion Paper | Discussion Paper | Discussion Paper

and a model shall be utilised. The algorithm employed is the GOME2/MetopA European Space Agency, ESA, Climate Change Initiative project, Ozone-CCI, GODFIT (GOME-type Direct FITting) version 3 algorithm (Lerot et al., 2014) whereas the model results originate from the European Centre for Medium-Range Weather Forecasts (ECMWF) repository at http://www.ecmwf.int. As part of the ESA Ozone_cci project, the GODFIT_v3 algorithm has been applied among others to the GOME2/MetopA, hereafter GOME2A, observations and the global validation of the GOME2A TOCs between 2007 and 2014 shall be used as an example for the possibilities of this type of post-processing improvement.

In Sect. 2.1 the Dobson spectrophotometer is briefly introduced, in Sect. 2.2 the GOME2/MetopA GODFIT_v3 algorithm is discussed, in Sect. 2.3 the application of the two effective temperatures on the Dobson TOCs is explained, as well as their comparison to auxiliary in situ-derived data. In Sect. 3 the results are analyzed and main conclusions follow in Sect. 4.

## 2    Data and methodology

### 2.1    The Dobson spectrophotometer total ozone columns

The Dobson instrument is a double monochromator with a dispersing spectrometer and a recombining spectrometer (Dobson, 1957a, 1958b). Consisting of a double prism monochromator, it is designed to measure the differential absorption in the UV region where $O_3$ absorbs strongly. Thus, the difference of intensities of the wavelengths, and not the absolute intensities of the single wavelengths, is measured by Dobson spectrophotometers. A discussion of the different error sources for the total ozone measurements with the Dobson instrument is given by Basher (1982), who concludes that with a well calibrated Dobson instrument the error on individual total ozone measurements may be estimated to be 2–3 %, later updated in Staehelin et al. (2003).

**AMTD**

doi:10.5194/amt-2015-343

**The impact of the ozone effective temperature**

M. E. Koukouli et al.

A continuously-updated selection of the Dobson instruments reporting data to the World Ozone and Ultraviolet Data Centre (WOUDC) at Toronto, Canada has already been used in the validation of different satellite TOC products such as in the works of Balis et al. (2007b), Antón et al. (2009), Loyola et al. (2011), Koukouli et al. (2012), Labow et al. (2013), Bak et al. (2015) among others. The station selection investigation and criteria have been explained in detail in Balis et al. (2007a, b) and, naturally, a continuous update of the in-house quality assurance of the chosen WOUDC stations is performed annually.

In this study, direct sun daily mean TOC values reported by 53 Dobson stations around the globe have been used as the validation standard; 19 of those are located in the Southern Hemisphere and 34 in the Northern Hemisphere. Out of those stations, 7 also host a Brewer spectrophotometer. The intercomparison between the TOCs reported by a Brewer and a Dobson instrument located in the same site often proves to be a useful tool as there exists a seasonality in the Brewer–Dobson against satellite differences investigated also in the past (see Fig. 1, as well as de Backer and De Muer, 1991; Vaníček, 2006; van Roozendael et al., 2008; Scarnato et al., 2010). These comparisons can prove to be a useful tool when assessing the temperature dependence of the Dobson absorption coefficients since the Brewer wavelengths were chosen so that stratospheric temperature changes would have the least effect on their reported TOCs (Kerr, 2002).

Dobson measurements are based on the use of effective ozone absorption coefficients that are derived for standard profiles of ozone and temperature, representative for the stratosphere where the bulk of the ozone absorption occurs. Note that the temperature dependence leads to a reduction in absorption – hence, an increase in observed ozone abundance – at colder than standard temperatures. Komhyr et al. (1993) give the temperature dependence for the Dobson TOCs of $0.13\%\,°C^{-1}$ at the $-45\,°C$ level, which has recently been verified by the work of Redondas et al. (2014). The operational Dobson algorithm assumes that the ozone absorption coefficients relate to a stratospheric temperature equal to $-46.3\,°C$ at all seasons and latitudes.

**AMTD**

doi:10.5194/amt-2015-343

**The impact of the ozone effective temperature**

M. E. Koukouli et al.

## 2.2 The GOME2/MetopA GODFIT_v3 total ozone columns

Within the ESA Ozone-CCI project, total ozone column records from GOME2A have been reprocessed with GODFIT version 3 (Lerot et al., 2014). This algorithm is an evolution of the retrieval baseline implemented in the operational GOME Data Processor v5 (Van Roozendael et al., 2012) and is based on the direct-fitting of simulated Huggins bands reflectances to the GOME2A observations. The GODFIT_v3 data products include for every satellite pixel, in addition to the retrieved total ozone column, a set of auxiliary parameters among which is the effective temperature. This temperature has been computed with the a priori temperature and ozone profiles used to simulate the reflectances. The GODFIT_v3 GOME2A TOCs, as well as those of GOME/ERS2 and SCIAMACHY/Envisat, have been evaluated on a global scale against Brewer and Dobson spectrophotometer TOCs (Koukouli et al., 2015). The mean bias to the ground-based observations is found to be within the ±1% level for all three sensors while the excellent decadal stability of the total ozone columns provided by the three European instruments falls well within the ESA Ozone-CCI project 1–3% requirement (van der A et al., 2011).

In Fig. 1 two examples of the validation process shown in Koukouli et al., 2015, are given. The monthly mean differences between GOME2A and ground TOCs are given for the Brewer (blue line) and the Dobson (red line) instruments located in Hradec Kralove, Czech Republic (left panel) and Hohenpeissenberg, Germany, (right panel). The mean difference and associated 1-sigma standard deviation are also given as insert. The comparisons are quite good for these two northern middle-latitude stations with differences well within the ±1% level. Since these are coincident measurements however in the same location, one would expect both types of ground-based instrument to show exactly the same behavior against the satellite sensor. The larger differences are found for the winter months as expected due to the fact that the effective temperature in the NH middle-latitudes deviates from −46.3 °C that is used in the standard

Discussion Paper | Discussion Paper | Discussion Paper | Discussion Paper |

# AMTD

doi:10.5194/amt-2015-343

**The impact of the ozone effective temperature**

M. E. Koukouli et al.

Dobson ozone retrieval algorithm. In the following section, the extend of this deviation and the magnitude of how it affects the TOCs is expanded upon.

## 2.3 The effective temperature dependency

A post-processing of the Dobson TOCs was performed in order to compensate for the well-known effective temperature dependency of the Dobson instruments (Staehelin et al., 2003). In reality, the absorption coefficients depend on temperature; as temperature changes depending on the season and the latitude, the absorption of solar radiation by ozone also changes. Therefore, for an accurate retrieval of TOC the actual temperature at all latitudes and seasons must be taken into account. However, the methodology of TOC retrieval from ground-based measurements does not allow partitioning of the ozone absorption at different atmospheric states. The Dobson instrument algorithm presumes that the stratospheric temperature is equal to −46.3 °C and the Brewer standard algorithm at −45 °C for all latitudes and seasons. Hence, ignoring this effect will lead to a seasonal dependent offset in the total ozone data (Fioletov et al., 2008; van der A, 2010).

The effective ozone temperature is defined as the integral over altitude of the ozone profile-weighted temperature and is derived by:

$$T_{\text{eff}} = \frac{\int_0^{\text{top}} T(z) \, O_3(z) \, dz}{\int_0^{\text{top}} O_3(z) \, dz} \tag{1}$$

Two different effective temperatures were investigated; one provided by the GOD-FIT_v3 algorithm, as discussed in Sect. 2.2, and one computed from the temperature and ozone profiles provided by a medium-range weather forecasting model by the ECMWF (European Centre for Medium-Range Weather Forecasts; http://www.ecmwf.int/) in order to produce new, post-corrected Dobson total ozone columns and compare them with the satellite TOC measurements. This ECMWF dataset was calculated from 6 hourly ECMWF temperature profiles extracted from the operational analyses, and

Discussion Paper | Discussion Paper | Discussion Paper | Discussion Paper | Discussion Paper |

**AMTD**

doi:10.5194/amt-2015-343

**The impact of the ozone effective temperature**

M. E. Koukouli et al.

**AMTD**

doi:10.5194/amt-2015-343

**The impact of the ozone effective temperature**

M. E. Koukouli et al.

the seasonally dependent Fortuin and Kelder ozone climatology (Fortuin and Kelder, 1998). For each ground station a dataset of daily values was created with the effective ozone temperatures interpolated to local noon (van der A et al., 2010).

The behaviour of these two effective temperature datasets was examined using as auxiliary data radiosonde and ozonesonde effective temperatures extracted from the WOUDC database. The criteria by which the selection of the ozonesonde stations was performed are firstly that a collocated Dobson instrument was needed, so as to perform direct comparisons of the effect of the ozonesonde effective temperature with those provided by ECMWF and GODFIT_v3. Secondly, we required global representativeness so as to examine the $T_{eff}$ behaviour of the different datasets at different latitudes. The ozonesonde ozone effective temperature was then calculated using the Eq. (1), with the integration being performed up to the balloon burst height, if that height exceeded the altitude of 30 km. Sondes that burst below 30 km were omitted from the calculations.

In Fig. 2, the effective temperatures are presented as time series for four Dobson locations around the globe: in the upper left plot, the temperatures over the Antarctic station in Syowa are shown; in the upper right, the tropical station in Samoa; in the lower left, a Northern middle latitude station in Hohenpeissenberg and in the lower right, an Arctic station in Ny Alesund. The ECMWF effective temperature is shown in blue, the GODFIT_v3 in red and the ozonesonde in green. All three methods seem to depict the seasonal variability quite satisfactorily and the slight bias between the ECMWF and the GODFIT_v3 $T_{eff}$s in the high Northern latitudes (lower right) is not worrysome. The mean values are also given in the figure, where the high standard deviation in the high latitude stations point to the seasonal variability of the atmospheric state in these latitudes. The correlation coefficients between GODFIT_v3 $T_{eff}$ and ECMWF $T_{eff}$, as well as those between GODFIT_v3 and Sonde, are given in Table I, where the details of the four representative Dobson locations are also shown. A very high correlation is found for the high and middle latitudes for both cases. The low correlation for the tropical case (Fig. 2, upper right) may be due to the very small seasonal variability,

Discussion Paper | Discussion Paper | Discussion Paper | Discussion Paper

testified by the low standard deviation of only around 2 °C. As a result, small variations between the $T_{eff}$s may cause these discrepancies, even though the mean values agree quite well. And exactly because the effective temperatures in the tropics is very close to the one actually used in the Dobson algorithm we do not expect those latitudes to be such an issue. The same behaviour was seen in other Dobson tropical stations examined as well [not shown here.] We hence feel confident that using the GODFIT_v3 calculated $T_{eff}$s, even though these are output from the same algorithm that produces the satellite TOCs, will not add any systematic bias in the comparisons with the ground-based TOCs.

## 3   Results and discussion

In order to post-process the Dobson total ozone columns the Eq. (2) was applied in order to calculate a new total ozone column, as per van der A et al. (2010):

$$O_{3\,new} = O_{3\,standard} \cdot \left[1 - 0.0013 \cdot (T_{eff\_new} - 226.7)\right] \tag{2}$$

where

1. $O_{3\,new}$ is the new ground total ozone column generated by using the new effective temperature,

2. $O_{3\,standard}$ is the retrieved total ozone column corresponding to the Dobson reference effective temperature ($-46.3$ °C),

3. 226.7 is the Dobson reference effective temperature expressed in Kelvin, and

4. $T_{eff\_new}$ is the effective temperatures derived from the GODFIT_v3 algorithm or the ECMWF database.

As a result of Eq. (2), two new post-processed ground-based TOCs exist and their inter-comparisons and effect on the original TOC are discussed below through Fig. 3.

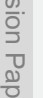

**AMTD**

doi:10.5194/amt-2015-343

**The impact of the ozone effective temperature**

M. E. Koukouli et al.

The comparisons, confined to the Dobson locations, have been averaged into six belts in 30° bins latitude and the following four are shown in Fig. 3 from left to right: the −90 to −60° S belt, the 0 to 30° N belt, the 30 to 60° N belt and the 60 to 90° N belt. In Fig. 3 upper row, the two temperatures are presented with the GODFIT_v3 shown in orange and the ECMWF in blue. As expected the higher variability is shown for the Antarctic (first panel) and the Arctic (fourth panel) with temperatures ranging between 200 and 240 K and 220 and 235 K respectively depending on the season. The NH tropical belt (second panel) shows almost negligible variability, well within 5 K, whereas a 10 K peak-to-peak for the NH middle latitudes (third panel) is found. Note that the SH tropical and middle latitude belts show exactly the same results, in reverse sign, and hence are omitted. As to the actual differences between the two temperatures, for the −90 to −60° S belt it is −0.28 ± 1.16 %; for the 0 to 30° N belt it is very similar at −0.27 ± 0.28 %; for the 30 to 60° N belt at −0.72 ± 0.20 % and for the 60 to 90° N belt, −0.97 ± 0.32 %.

In Fig. 3 lower row, the effect of post-correcting for the two effective temperatures on the Dobson TOCs is shown for the same latitude bands. The most prominent consequence is found for the Antarctic belt (first panel) with differences in the ozone values ranging from −1 to +3 % depending on the season, followed by the Arctic belt (fourth panel) with differences going from −1 to +1 % or even up to +2 % depending on the year. The effect is not as pronounced for the tropics (second panel) and the middle latitudes (third panel) where the differences go from −0.5 to +1 %, for the entire latitude band. The effect of the post-processed Dobson ozone observations on the validation of the GOME2A is given in subsequent Figures. To avoid repetitive discussion, only the GODFIT_v3 $T_{\text{eff}}$s will be utilised for the remainder of this paper.

In Fig. 4 the comparisons shown previously in Fig. 1 between the Dobson and Brewer TOCs located in the same station are updated using the post-processed Dobson TOCs, using the GODFIT_v3 effective temperature. The Dobson mean difference to the GOME2A observations has decreased from 1.15 ± 1.93 to 0.29 ± 1.32 % for the

**AMTD**

doi:10.5194/amt-2015-343

**The impact of the ozone effective temperature**

M. E. Koukouli et al.

Hradec-Kralove station and from $0.95 \pm 1.53$ to $0.16 \pm 1.08$ % for the Hohenpeissenberg Dobson, now bringing the two time series at precisely the same levels.

In Fig. 5 the nominal global validation of the GOME2A GODFIT_v3 dataset against collocated Dobson stations is shown in blue and is compared to post-processed Dobson data, in red. From the monthly mean percentage differences for the NH (upper left) and the SH (upper right) it is shown that the higher differences between ground and satellite decrease, whereas those monthly already hovering on the zero line remain unchanged. In numbers, the NH comparisons go from an original $0.63 \pm 0.66$ to $0.26 \pm 0.46$ % difference level and the SH comparisons go from $1.25 \pm 1.20$ to $0.80 \pm 0.71$ %. Most important is the fact that the known solar zenith angle dependency issue is more limited now, with the highest deviation decreasing from 2.38 to 1.37 % for the 80 to 85° bin, and near-zero dependency up to the 60 to 65° bin. The equivalent behaviour of the Brewer comparisons show the same near-zero dependency up to the 60 to 65° bin and the highest deviation of 2.34 % also for the 80 to 85° bin. However, a one-to-one comparison between Brewer and Dobson results is impossible due to the quite different geographical spread between the two sets of instruments. The expected improvement of the differences against the GODFIT_V3 effective temperature is shown in the bottom left panel of Fig. 5 where the dependency has all but disappeared and difference levels remain between 0 and 1 % for almost all temperatures examined. We hence conclude that, on a global scale, satellite-to-Dobson TOC comparisons benefit from this post-processing of the Dobson TOCs, as long as the $T_{\mathrm{eff}}$ employed has been independently validated against an independent source of measurements or modelling results.

## 4   Conclusions

In this paper, the impact of the total ozone effective temperature on satellite validation using the global Dobson spectrophotometer network was presented using the European Space Agency Ozone Climate Change Initiative GOME-type Direct FITting ver-

Discussion Paper | Discussion Paper | Discussion Paper | Discussion Paper |

**AMTD**

doi:10.5194/amt-2015-343

**The impact of the ozone effective temperature**

M. E. Koukouli et al.

sion 3 algorithm as it was applied to the GOME2/MetopA observations. Ozone effective temperatures calculated by the GODFIT_v3 algorithm, as well as the ones extracted from the European Centre for Medium-Range Weather Forecasts model, were examined and evaluated against collocated ozonesonde measurements. Both sets of effective temperatures were found to agree to a satisfactory degree to the in situ observed effective temperatures and also to result in the same effect on the Dobson total ozone columns. By applying a post-processing to the reported Dobson total ozone columns the comparisons to the GOME-2A GODFIT_v3 columns results in:

1. Examining select stations around the world that host both a Dobson and a Brewer instrument it was shown that for the Hradec-Kralove station in the Checz Republic the Dobson mean difference to the GOME2A observations has decreased from $1.15 \pm 1.93$ to $0.29 \pm 1.32$ %, and for the Hohenpeissenberg station, Germany, from $0.95 \pm 1.53$ to $0.16 \pm 1.08$ %. The equivalent Brewer statistics are $-0.04 \pm 1.17$ and $0.31 \pm 1.06$ % respectively.

2. NH comparisons improve from the $0.63 \pm 0.66$ to the $0.26 \pm 0.46$ % difference level and the SH comparisons go from $1.25 \pm 1.20$ to $0.80 \pm 0.71$ %. Comparisons to Dobson stations located in all latitude bands examined benefit from this post-correction.

3. The known solar zenith angle dependency in the satellite-Dobson TOC differences is much more limited now, with the highest deviation decreasing from 2.38 to 1.37 % for the 80 to 85° bin, and near-zero dependency up to the 60 to 65° bin. The equivalent behaviour of the Brewer comparisons show the same near-zero dependency up to the 60 to 65° bin and the highest deviation of 2.34 % also for the 80 to 85° bin.

4. Effective temperatures calculated by either the GODFIT_v3 satellite algorithm or the ECMWF model may be used for the post-processing of the Dobson total ozone columns.

**AMTD**

doi:10.5194/amt-2015-343

**The impact of the ozone effective temperature**

M. E. Koukouli et al.

Discussion Paper | Discussion Paper | Discussion Paper | Discussion Paper

We hence strongly recommend that any future global satellite total ozone validation activities using the standard Dobson ground-based total ozone measurements be performed using post-processed Dobson total ozone columns using Eq. (2) and quality assured effective temperature data.

*Acknowledgements.* The authors would like to warmly thank the scientists working with the Dobson and Brewer spectrophotometers and their continuous efforts to provide timely and quality assured total ozone columns to the WOUDC database. We would also like to warmly thank the scientists maintaining the WOUDC database for their continuous hard work and immediate response to data requests. We further wish to thank Mark Allard and Ronald van der A, from the Koninklijk Nederlands Meteorologisch Instituut (KNMI), De Bilt, the Netherlands, for providing us with the ECMWF effective temperature data set.

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

**Table 1.** The details from the four representative Dobson locations presented in Fig. 2 sorted in latitude. The correlations between GODFIT_v3 and ECMWF $T_{\mathrm{eff}}$, as well as between GODFIT_v3 and Sonde $T_{\mathrm{eff}}$, are given in columns six and seven, respectively.

| Station name | Country | Station number | Latitude | Longitude | Correlation $R^2$ between GODFIT_$T_{\mathrm{eff}}$ & ECMWF_$T_{\mathrm{eff}}$ | Correlation $R^2$ between GODFIT_$T_{\mathrm{eff}}$ & SONDE_$T_{\mathrm{eff}}$ |
|---|---|---|---|---|---|---|
| Ny_Alesund | Norway | 89 | 78.93 | 11.88 | 0.967 | 0.955 |
| Hohenpeissenberg | Germany | 99 | 47.8 | 11.02 | 0.971 | 0.952 |
| Samoa | USA | 191 | −14.25 | −170.57 | 0.543 | 0.490 |
| Syowa | Antarctica | 101 | −69.00 | 39.58 | 0.891 | 0.962 |

Discussion Paper | Discussion Paper | Discussion Paper | Discussion Paper |

**AMTD**

doi:10.5194/amt-2015-343

**The impact of the ozone effective temperature**

M. E. Koukouli et al.

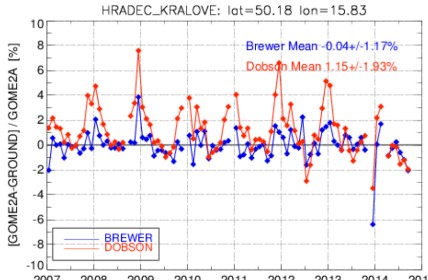
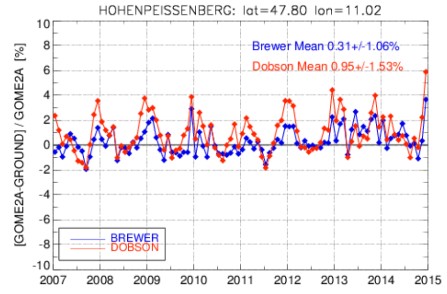

**Figure 1.** Monthly mean differences between GOME2A and Brewer (blue) and Dobson (red) total ozone columns for two middle latitude sites, in Hradec Kralove, Czech Republic (left panel) and Hohenpeissenberg, Germany, (right panel).

## AMTD

doi:10.5194/amt-2015-343

**The impact of the ozone effective temperature**

M. E. Koukouli et al.

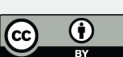

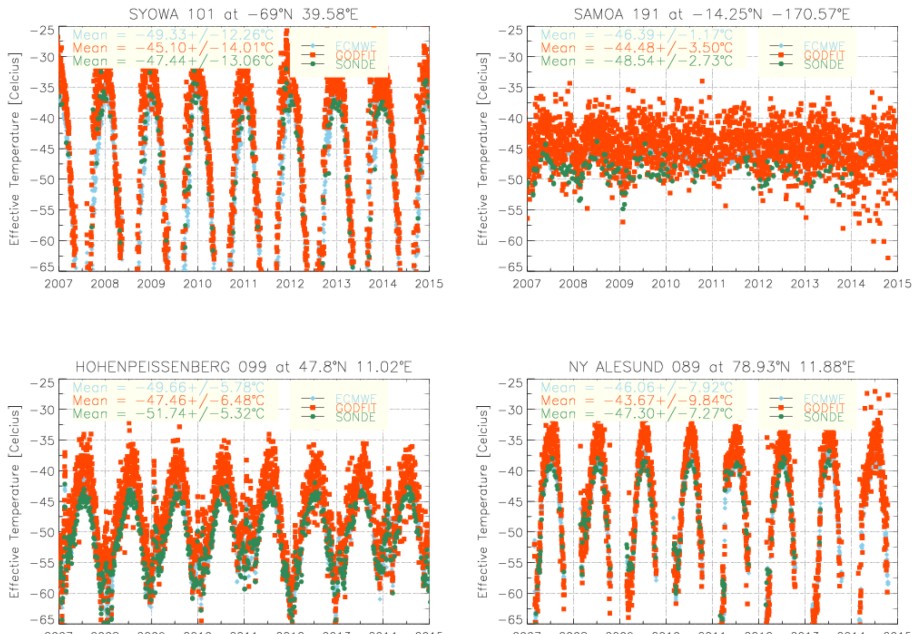

**Figure 2.** Time series of the effective temperatures estimated by ECMWF (blue), GODFIT_v3 (red) and ozonesondes (green) for four Dobson locations: upper left, an Antarctic station in Syowa; upper right, a tropical station in Samoa; lower left, a Northern middle latitude station in Hohenpeissenberg and lower right, an Arctic station in Ny Alesund. The mean values are also given in the upper left corner of each plot.

Discussion Paper | Discussion Paper | Discussion Paper | Discussion Paper

**AMTD**

doi:10.5194/amt-2015-343

**The impact of the ozone effective temperature**

M. E. Koukouli et al.

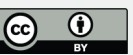

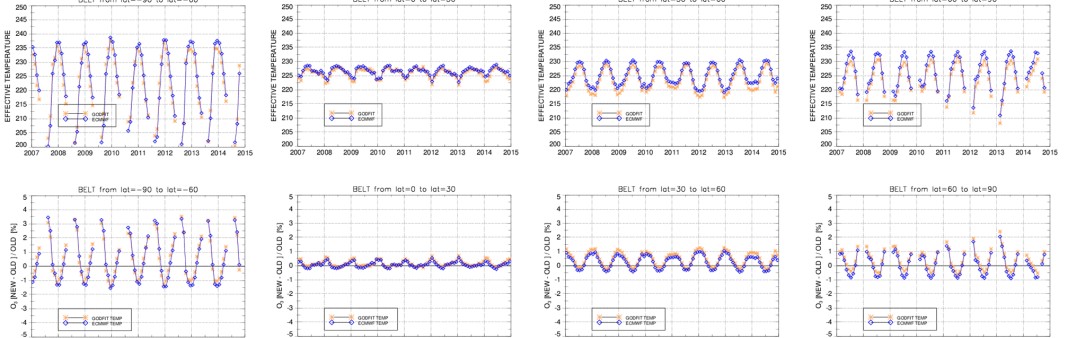

**Figure 3.** Upper row: Monthly mean time series of the effective temperature from the GOD-FIT_v3 algorithm (orange) and the ECMWF model (blue) for the Dobson locations. Lower row: The percentage difference between the nominal Dobson TOCs and the one calculated using the GODFIT_v3 algorithm (orange) and the ECMWF model (blue) for the Dobson locations. From left to right: the −90 to −60° S belt, the 0 to 30° N belt, the 30 to 60° N belt and the 60 to 90° N belt.

Discussion Paper | Discussion Paper | Discussion Paper | Discussion Paper

**AMTD**

doi:10.5194/amt-2015-343

**The impact of the ozone effective temperature**

M. E. Koukouli et al.

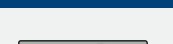

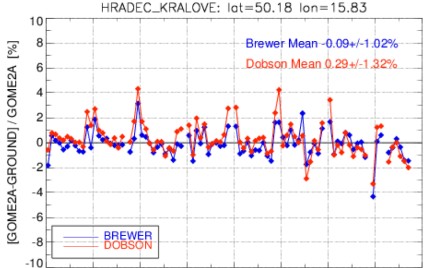
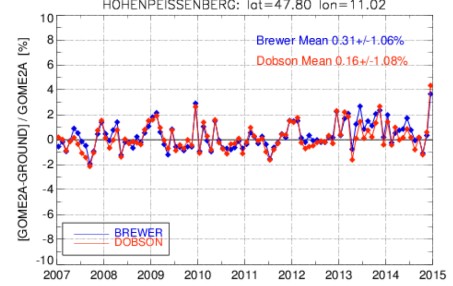

**Figure 4.** Same as Fig. 1 with the Dobson TOCs being post-processed using the GODFIT_v3 effective temperature.

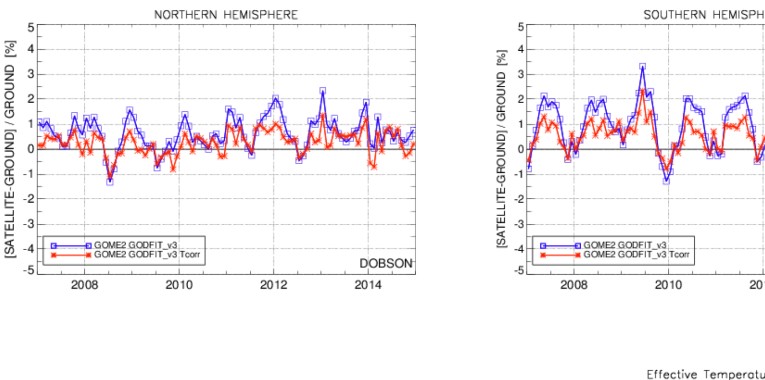

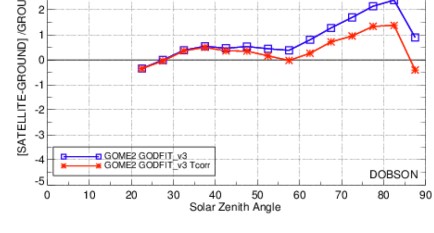

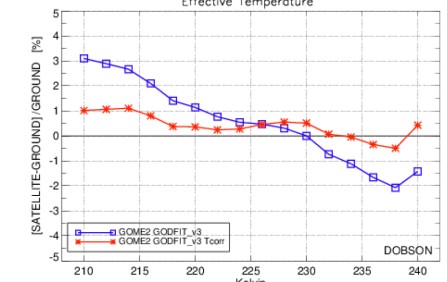

**Figure 5.** Global comparisons between the nominal (blue) and the post-processed (red) Dobson and GOME2 GODFIT_v3 TOCs. Upper row: the monthly mean time series for the NH (left) and the SH (right) Dobson stations. Bottom row left: the solar zenith angle dependency. Bottom row right: the effective temperature dependency.

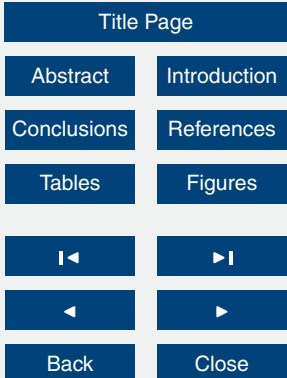

Discussion Paper | Discussion Paper | Discussion Paper | Discussion Paper |

**AMTD**

doi:10.5194/amt-2015-343

**The impact of the ozone effective temperature**

M. E. Koukouli et al.