# Peer review of "The impact of the ozone effective temperature on satellite validation using the Dobson spectrophotometer network"

_Atmospheric Measurement Techniques, 2015_

## Short Comment (SC1) · 23 Jan 2016

A referee has already mentioned the fact that no clear statement exists in the text about which ozone cross-section set(s) are used in the calculations. I'd like to add to this that every time I see a good agreement between satellite and Dobson/Brewer or an improved agreement after some correction I always wonder how Bass&Paur for Dobson/Brewer, which we know is internally inconsistent (especially in regard to T dependence), can produce a good agreement with some other cross-section set that the satellite community use. In fact, different satellite platforms potentially use different cross-sections. Also, the formula that is used for correcting the Dobson data is attributed to van der A. et.al (2010), but in that paper the authors simply refer to Kerr

(2002) for this correction and Kerr's paper only mentions this correction in a discussion section as a possibility with several "if" statements.

---

## Referee Comment (RC2) · Anonymous Referee #1 · 11 Feb 2016

**The impact of the ozone effective temperature on satellite validation using the Dobson spectrophotometer network**

M. E. Koukouli, M. Zara, C. Lerot, K. Fragkos, D. S. Balis, and M. van Roozendael

February 11, 2016

**1 Summary**

The paper gives a useful evaluation of how the application of the know effective temperature dependence on Dobson spectrometer improves the satellite ground base comparison. The authors validate the $T_{eff}$ calculations derived from satellite retrieval and with the ECWMF weather model based Temperature profiles with the same a priori climatology used from the satellite retrieval with ozonesonde based derivation.

The improvement of the comparison is clear which suggest to implement an operative implementation for Dobson ozone observations

**2 Comments**

As other referee comment a mention of the different ozone cross section used by satellite retrieval and Brewer /Dobson will be mentioned.

Page 5 5 Some references are missing on the bibliography (Anton and Labow)

Page 7 25 Can you give more details about the operative analysis used.

Page 8 10 Do you have an estimation of the bias on $T_eff$ calculated by ozonesondes due the fact of limited altitude of the ozonesonde.

Page 9 10, I think This formula was used from first time by Roozendael 1998 (Formula 4)

Page 11 10 : The "know solar zenith angle" dependence should be described and or referenced.

Page 11 10: The brewer comparison are not shown.

Table 1: I suggest to add also the ECWMF vs SONDE stats and include the mean values of the figure 2.

Figure 2: Plotting the differences to the ozonesonde rather the series could be more illustrative of the comparison.

Figure 5 : The lower panel a description of how the zenith angle and temperature dependence are calculated.

---

## Author Comment (AC1) · 5 Apr 2016

Referee comment of the manuscript:

AMT-2015-343: The impact of the ozone effective temperature on satellite validation using the Dobson spectrophotometer network - M.E. Koukouli, M. Zara, C. Lerot, K. Fragkos, D. S. Balis, and M. Van Roozendael

**General comments:**

The content of the paper is a very interesting contribution to the issue of data quality in the Dobson spectrophotometer network. The presentation is well structured, thus it is no problem to understand and to follow the intention of the paper and the results of the proposed improvements.

*We would like to warmly thank referee #2 for his/her valuable comments towards the improvement of our paper.*

**Specific comments:**

i.      Minor Issues:
Solar zenith angle dependency often mentioned in the text as known, but no possible explanation given for the remaining dependency after Teff-correction.
*A relevant paragraph was added in the text.*

Replace "early 1980s" by "late late 1970s" on page 2, line 11: TOMS on Nimbus 7 already starts measurements in 1978.

*Thank you for noticing this, a small typo was all it was.*

Replace "Since year 1958" by "Since 1957/1958" on page 2, line 15. Reference "Brönnimann et al, (=Staehelin, Farmer, Cain, Svendby and Svenoe), Total ozone observations prior to the IGY I: A history, Q.J.R. Meteorol. Soc. 2003 as related reference would be nice.

*Thank you for bringing this very informative reference to our attention, added as requested.*

The effect of the Teff-is described as seasonal several times (e.g. on page 6, line 4 – 5), but it can also effect the ozone observations on a daily base (rapid, intense change of weather situation). This time scale should be mentioned too.

*A comment to that effect, as well as a reference, was added to the text.*

As one example of time series Ny Alesund is shown (page 8). Fortunately only the agreement of the various Teffs in the annual course is shown. However, Thule (Dobson operation recently ceased) or Reykjavik (still active) might be better stations, as Dobson No. 008 at Spitsbergen has been out of operation since many years

*For this part of the study, we were mostly concerned with locating stations with a long, as well as gap-free, time series of ozonesonde data. No ozonesonde fly out of Reykjavik or Thule,*

*to the best of our knowledge. Both the NDACC and the WOUDC repositories were searched for an appropriate, typical, Arctic location and we agreed that Ny Alesund fits the bill. We hence consider that this station may be representative of the region and hence provide the ozonsonde vs ECMWF vs satellite algorithm comparison.*

ii.      Major issues:

It is nowhere mentioned that the Bass/Paur absorption coefficients are still in use in the Dobson and Brewer Spectrometer algotihms. Planned introduction of ozone new cross sections/absorption coefficients (University Bremen) might change the results. Redondas et al is already mentioned, but not in this context.

*A relevant section discussing the different absorption cross-sections and their reported effective temperature dependence was added in the text.*

It is also nowhere mentioned which ozone cross sections are used in the satellite algorithms (old???, already new, but not Uni Bremen???).

*The relevant information was added in the text.*

Explanation of the Teff effect on ozone values (page 4, lines 19 to 21) is a little bit confusing. Colder temperatures cause reduction in real absorption coefficents, which would give increased ozone. The Dobson however still uses larger absorption coefficients, thus the observed Dobson ozone values are lower and this causes the annual pattern in the Dobson- Brewer difference.

*Text updated as requested.*

Technical corrections:

References:
- o   Anton et al 2009 is missing in references(cited on page 4, line 2)
- o   Labow et al 2013 is missing in references (cited on page 4, line 2)
- o   van Roozendael et al. 2008 is missing in references (cited on page 4, line 12/13, van roozendael et al. 1998 in references?)

*References added and corrected.*

Is bottom left panel of Figure 5 (on page 12, line 17 and page 13) not bottom right panel of Figure 5?.

*Reference corrected.*

---

## Author Comment (AC2) · 5 Apr 2016

The paper gives a useful evaluation of how the application of the know effective temperature dependence on Dobson spectrometer improves the satellite ground base comparison. The authors validate the Tef f calculations derived from satellite retrieval and with the ECWMF weather model based Temperature profiles with the same a priori climatology used from the satellite retrieval with ozonesonde based derivation.

The improvement of the comparison is clear which suggest to implement an operative implementation for Dobson ozone observations

*We would like to warmly thank referee #1 for his/her valuable comments towards the improvement of our paper.*

As other referee comment a mention of the different ozone cross section used by satellite retrieval and Brewer /Dobson will be mentioned.

*We agree and apologize for this small omission. The relevant paragraph has been added in the text.*

Page 5 5 Some references are missing on the bibliography (Anton and Labow)

*References added.*

Page 7 25 Can you give more details about the operative analysis used.

*A line to that affect was added in the text.*

Page 8 10 Do you have an estimation of the bias on Tef f calculated by ozonesondes due the fact of limited altitude of the ozonesonde.

*We have added a relevant analysis, with a new Figure and statistics in the relevant section.*

Page 9 10, I think This formula was used from first time by Roozendael 1998 (Formula4)

*Reference added in the text.*

Page 11 10 : The "know solar zenith angle" dependence should be described and or referenced.

*A relevant paragraph was added in the text.*

Page 11 10: The brewer comparison are not shown.

*A relevant mention to this fact was included.*

Table 1:  I suggest to add also the ECWMF vs SONDE stats and include the mean values of the figure 2.

*The requested statistics were added to the table.*

Figure 2: Plotting the differences to the ozonesonde rather the series could be more illustrative of the comparison.

*Indeed, plotting differences is illustrative for most cases, however since the differences here are so small the Figures become too crowded and the seasonality effect is not as evident. We have added more statistics in the Table and we hope that this will cover any remaining questions.*

Figure 5 : The lower panel a description of how the zenith angle and temperature dependence are calculated.

*The relevant information was added in the text.*

---

## Author Comment (AC3) · 5 Apr 2016

**V. Savastiouk vl@dimir.ca Received and published: 23 January 2016**

A referee has already mentioned the fact that no clear statement exists in the text about which ozone cross-section set(s) are used in the calculations. I'd like to add to this that every time I see a good agreement between satellite and Dobson/Brewer or an improved agreement after some correction I always wonder how Bass&Paur for Dobson/Brewer, which we know is internally inconsistent (especially in regard to T dependence), can produce a good agreement with some other cross-section set that the satellite community use. In fact, different satellite platforms potentially use different cross-sections. Also, the formula that is used for correcting the Dobson data is attributed to van der A. et.al (2010), but in that paper the authors simply refer to Kerr (2002) for this correction and Kerr's paper only mentions this correction in a discussion section as a possibility with several "if" statements.

**Response by M. E. Koukouli and co-authors.**

mariliza@auth.gr Tuesday, April 05, 2016

We would like to thank Dr Sevastiouk for making this important distinction. In the revised version, it is now made clear which ozone cross-section data set is used in the various data sets inter-compared and validated. Furthermore, a discussion on the effective temperature dependency on each of the ozone cross-section data set was added. Concerning the comparisons between satellite and ground-based data these mostly aim to examine the consistency between the two sets of observations; it is a true fact that there could be differences not only in the cross sections employed but also in the algorithm principles followed by each type of instrument. Traditionally, when comparing satellite to ground TOCs, we try to identify the reasons for good or bad agreement, and when changes are applied to the satellite algorithms we can quantify the effect and the sign of these changes. In this work, we have turned the case on its head, as it may; i.e. we introduce a documented change in the groundbased TOC and discuss its effect on the comparisons between ground and satellite. Of course all this work is limited by known, as well as unknown, uncertainties on both platforms (ground-based and satellite). Finally, the corrective equation we employed is the one used by van Roozendael et al., 1998 as well as van der A et al., 2010, we did not mean to imply anything on its provenance.